# The Impact of Corporate Social Responsibility on Financial Performance and Brand Value

**Jing Zhang**  **and Ziyang Liu** *

Graduate School, Kyonggi University, Suwon 16227, Republic of Korea; zhangjing@kgu.ac.kr
* Correspondence: victor@kgu.ac.kr

**Abstract:** In recent years, there has been an increasing amount of theoretical research on corporate social responsibility and its influence on practical activities. The impact of corporate social responsibility on business performance has received attention from scholars and managers. However, the existing research lacks the empirical analysis concerning the moderating effects of long-term business performance (brand value) and social capital. This study was based on the relevant data from listed, Chinese companies and conducted regression analysis on the impact of corporate social responsibility on financial performance and brand value, exploring its moderating effects under different social capital. The results showed that Corporate Social Responsibility (CSR) was significantly positively correlated with financial performance and brand value. Both horizontal and vertical social capital played a positive moderating role in the impact of CSR on financial performance and brand value. These conclusions differed between companies that were required to disclose and those that had voluntarily disclosed, as well as between heavily polluting industries and non-heavily polluting industries. This article enriches the existing theoretical framework and provides decision-making references for business managers on whether to take on corporate social responsibility, contributing to the theoretical understanding of corporate sustainable development from a social responsibility perspective.

**Keywords:** corporate social responsibility; enterprise performance; brand value; sustainable development of enterprises; horizontal social capital; vertical social capital

## 1. Introduction

In 1924, management philosopher Oliver Sheldon proposed the concept of corporate social responsibility (CSR) and advocated that "serving society is the fundamental driving force and foundation of industrial development" [1,2]. Under this theory, there has been an increasing amount of theoretical research and practical activities related to CSR, indicating the importance of CSR for a company's development. In China, the 2008 Sanlu milk scandal marked the beginning of the concept of corporate social responsibility entering the public eye. Subsequent incidents of corporate social responsibility failures have made people realize that the public should supervise companies in fulfilling their social responsibilities: Baidu, China's largest search engine, promoted Putian hospitals, indirectly leading to the death of cancer patient Wei Zexi; Changchun Changsheng illegally produced freeze-dried human rabies vaccines that almost entered the market, resulting in a fine of 9.1 billion yuan; Ofo failed to refund user deposits due to a broken funding chain, with a waiting list of up to 12 million people. However, there have also been companies that have fulfilled their social responsibilities and deserve recognition: China Ocean Shipping(Group) Company donated a total of CNY 310 million in major natural disasters, such as the southern rain and snowstorms, the Wenchuan earthquake in Sichuan, and the Yushu earthquake in Qinghai, and has received the China Charity Award from the Ministry of Civil Affairs multiple times; in 2016, China Huaneng Group released a greenhouse gas emissions report, systematically demonstrating new methods, new ideas, and new achievements in low-carbon emission reduction work; in 2018, Yili released a biodiversity protection annual report globally, systematically disclosing efforts to promote the fulfillment of social responsibilities throughout

the industry chain and carry out multiple biodiversity practice projects. From the above examples, it was evident that there were various reasons for companies to fulfill their social responsibilities, while the reasons for companies not fulfilling their social responsibilities have often been to maximize profits. Companies are economic organizations designed to achieve business performance. Therefore, will fulfilling their social responsibilities provide value losses to companies? And what is the relationship between corporate social responsibility and business performance?

The relationship between corporate social responsibility and business performance, whether for corporate managers, shareholders, or stakeholders, is immeasurable in value. The stakeholder theory suggested that corporate social responsibility was positively correlated with financial performance, as fulfilling social responsibilities could enhance stakeholder satisfaction, ultimately leading to better financial performance. Conversely, failing to meet the expectations of various stakeholders would generate market fears and ultimately result in lost profit opportunities.

There has been no unified conclusion among domestic and foreign scholars regarding the relationship between corporate social responsibility and financial performance. This may be due to the inconsistent evaluation methods for business performance, the different indicators set, and the differences in sample selection, leading to a potentially contradictory situation. Through a review of past literature, this study found that existing research on the impact of corporate social responsibility on financial performance has often focused on its short-term effects (financial performance) and has lacked attention to long-term performance (brand value). In fact, in some cases, the long-term benefits of corporate performance (brand value) has far outweighed the short-term business performance (financial performance) [3–6]. Additionally, existing research on the impact of corporate social responsibility on business performance has primarily focused on developed countries and overlooked developing countries. The institutional culture of different countries can affect corporate social responsibility practices, as institutional conditions can alter the benefits and losses of a company's actions, thereby influencing a company's motivation and decisions. Therefore, whether there are different conclusions regarding the impact of corporate social responsibility on financial performance in emerging economies remains an unsolved mystery.

Based on this, this study, using data from listed Chinese companies, divided business performance into short-term performance (financial performance) and long-term performance (brand value), and explored the differentiated impact of corporate social responsibility on each. Additionally, this study also explored the moderating role of social capital in this relationship. As an important resource for companies, social capital provides benefits to companies through social networks or reciprocal behavior. Companies with different social capital receive different feedback when fulfilling social responsibilities. In additional research, this study divided the sample into companies that voluntarily disclosed their social responsibility reports and those that were mandated to disclose them, as well as heavy-polluting companies and non-heavy-polluting companies, to explore the differences in the research results in different subgroups.

The main contribution of this study relied on exploring the impact of CSR on both short-term performance (i.e., financial performance) and long-term performance (i.e., brand value), enriching existing research, and providing references for companies on whether to fulfill CSR. Additionally, the study also investigated the different roles of various social capital in the relationship between CSR and corporate performance as well as brand value. Finally, the study further analyzed the heterogeneity of the proactive and passive information disclosures by companies, as well as companies in heavy-polluting industries and non-heavy-polluting industries, in order to provide more detailed recommendations for companies when fulfilling CSR.

## 2. Conceptual Model and Research Assumptions

*2.1. Conceptual Model*

The purpose of this study was to discuss whether fulfilling CSR could have a positive impact on corporate performance and, therefore, divided corporate performance into financial performance and brand value. Financial performance was used to represent the impact of CSR on short-term corporate performance, while brand value was used to represent the impact of CSR on long-term corporate performance. In addition, this study also explored the moderating role of social capital in this relationship. Social capital provided benefits to companies through social networks or reciprocal behavior. Companies with different social capital would receive different feedback when fulfilling their social responsibilities. This model measured the moderating role of social capital from the perspectives of vertical social capital and horizontal social capital.

At the same time, this study introduced other control variables when examining the relationship between social capital and financial performance as well as brand value, such as firm size, ownership nature, managerial competence, debt-paying ability, market competition, advertising intensity, years of listing, management ownership ratio, and fixed asset ratio. This enabled the final results to be more in line with theoretical reality. The model framework is shown in Figure 1.

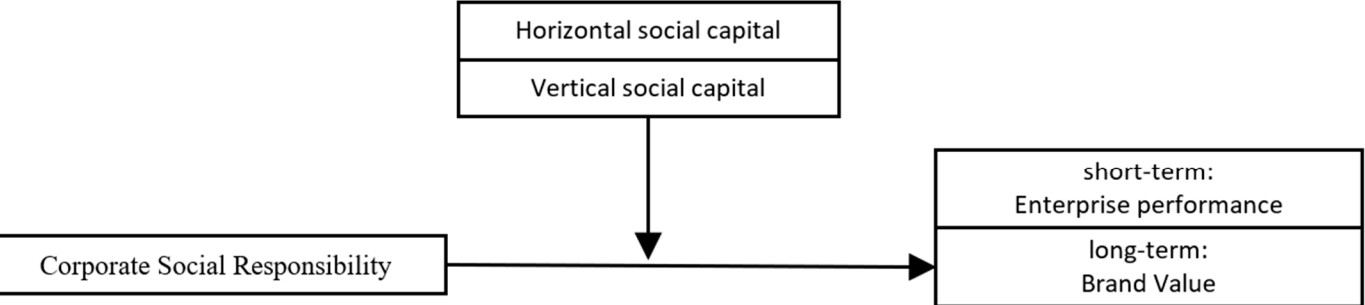

**Figure 1.** CSR and conceptual model of enterprise financial performance.

*2.2. Research Assumptions*

2.2.1. The Impact of Corporate Social Responsibility on Financial Performance

Corporate social responsibility referred to the moral obligations that organizations undertake internally and towards society, such as environmental protection and safeguarding employee rights, while ensuring the interests of the majority of shareholders [7–10]. Financial performance referred to the financial results achieved by an organization after a certain period of operation through the implementation of management strategies formulated by the management team, typically reflected through accounting indicators [11–13]. This study argued that there was a positive correlation between corporate social responsibility and financial performance.

Firstly, according to signal theory, corporate social responsibility was a mechanism for transmitting signals. As companies are a set of contracts between various stakeholders, there are various principal–agent relationships in the execution of these contracts, such as between shareholders and managers, between creditors and company managers or shareholders, and between employees, suppliers, customers, and managers or shareholders [14–16]. However, there was asymmetrical information in the development process of companies, and stakeholders were not clear about the actual situation of the company, which could lead to high agency costs between companies and stakeholders, thus affecting the development of companies. In order to solve the problem of asymmetrical information, companies had to transmit certain signals to stakeholders to indicate that they were trustworthy. Corporate social responsibility was such a mechanism for transmitting signals. It helped companies win the trust and support of stakeholders, maintain long-term cooperative relationships with stakeholders, and achieve sustainable development.

Secondly, according to stakeholder theory, corporate social responsibility was a mechanism for implementing transactions. Since companies were considered a set of contracts between various stakeholders, these contracts defined the rights and responsibilities of each party. In order for companies and stakeholders to mutually obtain resources and a good operating environment, they had to assume corresponding responsibilities and mutually provide resources and a good operating environment. It could be said that the contractual relationship between companies and stakeholders was actually a transactional relationship. Corporate social responsibility had to maximize the quality and efficiency of this transaction, thereby ensuring that resources and a good operating environment could be obtained from stakeholders [17–20]. Conversely, if companies did not attach importance to social responsibility and could maintain good transactional relationships with stakeholders, they could obtain resources and support from stakeholders, and may even have to bear corresponding risks, such as reputation damage, increased transaction costs, consumer resistance, talent loss, difficult refinancing, and legal sanctions.

Thirdly, corporate social responsibility was a mechanism for creating value. Corporate social responsibility ensured the interests of stakeholders, created value for the entire society, and at the same time, created value for itself. A large number of studies have shown that companies could improve their reputation, enhance brand loyalty, improve employee efficiency, reduce financing costs, and improve their relationship with government regulatory agencies by assuming social responsibility, thereby reducing business risks and creating greater commercial value [21–24]. Therefore, it could be said that corporate social responsibility was not a simple altruism, but a win–win mechanism of both altruism and self-interest, which could provide long-term financial benefits to companies and achieve their sustainable development goals.

Therefore, this study proposed the following hypothesis:

**H$_1$**. *CSR has a positive impact on financial performance.*

2.2.2. The Impact of Corporate Social Responsibility on Brand Value

Brand value referred to the value that a brand provided to both the company and consumers after establishing a relationship, and it was an intangible asset. This study argued that there was a positive correlation between fulfilling social responsibility and brand value.

Fulfilling corporate social responsibility promoted brand value from multiple perspectives. Firstly, from an external perspective, actively assuming social responsibility reflected a company's good sense of social ethics, demonstrated a positive image, and showcased the power of corporate culture, which thereby enhanced consumer brand identification, improved corporate reputation, and generated positive word-of-mouth [25–28]. Secondly, by providing superior products and services, a company could establish a good interactive relationship with consumers, attract the attention of modern customers who care about social development and guide them to choosing the brand while maximizing stakeholder satisfaction [29–31].

Secondly, within the company, by respecting employee individuality and creating a platform for career development, the company could provide outstanding talents with broad development opportunities; safeguard basic employee labor rights and interests; attract and nurture more outstanding talent; enhance labor efficiency; and create a favorable employment atmosphere and talent brand within the company [32–34]. For suppliers and distributors at both ends of the industrial chain, the company could build harmonious cooperative relationships based on the principles of mutual benefit; standardized, institutionalized, and humanized development; honesty and trustworthiness; continuously improving the company's management; perfecting the company's management system and business methods; establishing broad channels; attracting more partners; continuously expanding the company's development scale; and creating high-end, high-quality products, technology, and service brands, thereby improving the company's economic benefits [35,36]. Therefore, it could be observed that actively fulfilling their social responsibility from both

internal and external aspects contributed towards improving the company's economic benefits by constructing an efficient enterprise brand-value system and promoting the sustainable and healthy development of the company. Therefore, this study proposed the following hypothesis:

**H₂.** *CSR has a positive impact on brand value.*

### 2.2.3. The Regulatory Role of Horizontal Social Capital

Horizontal social capital referred to the connections that a company had with investors, suppliers, retailers, and customers, among others. This study argued that, when horizontal social capital was at a high level, fulfilling social responsibility had a greater impact on financial performance and brand value.

Firstly, horizontal social capital allowed companies to gain unique information that could not be obtained through conventional means. This could provide valuable insights for the company's decision-making and strategic choices in fulfilling its social responsibility, resulting in positive feedback. Secondly, companies with higher levels of horizontal social capital were more likely to communicate and exchange technical and operational experiences with other equal business entities within the network. As compared to companies with lower levels of horizontal social capital, they could gain unique competitive advantages. Lastly, due to the information exchange resulting from the aforementioned two points, companies would build trust and reputation among social-network relationships and business partners. By effectively utilizing this strategic resource, they could achieve greater economic benefits and brand value, as compared to companies with lower levels of horizontal social capital. Therefore, this study proposed the following hypothesis:

**H₃ₐ.** *Horizontal social capital positively moderates the relationship between CSR and financial performance.*

**H₃ᵦ.** *Horizontal social capital positively moderates the relationship between CSR and brand value.*

### 2.2.4. The Regulatory Role of Vertical Social Capital

Vertical social capital referred to the social connections that a company had with government institutions and officials. This study argued that when a company's vertical social capital was at a high level, fulfilling social responsibility had a greater impact on financial performance. As a socialist country, China has vigorously promoted the market economy since the Chinese reform and opening-up policy. However, government macroeconomic policies still play a crucial role in influencing market operations. Therefore, establishing harmonious and friendly social networks with the government was essential, and vertical social capital played a moderating role. The reasons were as follows:

Firstly, companies with high levels of vertical social capital could mitigate political and economic risks by maintaining stable government relationships when facing unfavorable situations. Secondly, the industry guidance policies introduced by the government had a significant impact on companies and having a higher level of vertical social capital allowed companies to fully utilize asymmetrical information.

Additionally, the government controlled a considerable amount of exclusive resources, such as policy subsidies and land-use rights. Having a harmonious and positive government relationship network could facilitate the companies' access to resources, thereby enhancing their competitive advantage. All these factors strengthen the positive impact of fulfilling social responsibility on financial performance and brand value. Therefore, this study proposed the following hypothesis:

**H₄ₐ.** *Vertical social capital positively moderates the20 relationship between CSR and financial performance.*

**H₄ᵦ.** *Vertical social capital positively moderates the relationship between CSR and brand value.*

## 3. Model Setting and Data Selection

### 3.1. Data Sources

The main source of the sample in this study was the A-share listed companies on the Shanghai and Shenzhen stock exchanges from 2013 to 2022. To ensure a more relevant study, the research data underwent the following treatments:

(1) This study selected companies that had been selected for the Top 500 Most Valuable Chinese Brands for 10 consecutive years as the data sample. This was based on two points: firstly, one of the dependent variables in this study was brand value, and the measurement of brand value in this study was primarily based on the Top 500 Most Valuable Chinese Brands, so the sample companies needed to be on the list. Secondly, in the selection of samples, we chose companies that had been on the list for 10 consecutive years. This was because in the analysis of this study, the company's performance was divided into two parts, short-term performance (financial performance) and long-term performance (brand value), and brand value required a certain historical consistency. As compared to companies that were not consistently on the list over a 10-year period, companies that had been on the list for 10 consecutive years had more stable and reliable formation and changes in brand value. These companies may have had a positive impact on their brand image and value through the active fulfillment of their corporate social responsibility during their long-term development process.

(2) Financial and insurance companies were excluded due to their different financial statement structures, major accounting items, and business models, as compared to companies in other industries.

(3) To ensure data consistency and stability, the stocks in abnormal trading states, such as ST, SB, and PT, were excluded from the sample.

In the end, the panel data from 81 companies spanning 10 years were collected, resulting in a total of 810 observations. The data on corporate social responsibility (CSR) were obtained from the HEXUN database and RKS database, while other data were sourced from the Guotai An database.

### 3.2. Variable Setting

#### 3.2.1. Dependent Variables

Financial Performance: Considering the objectivity and accuracy of the data, this study used accounting-based indicators to measure financial performance. The return-on-assets (ROA) was selected as the indicator to measure the financial performance of companies.

Brand Value: From an accounting perspective, the existence of a brand enabled a company to obtain a higher present value of future cash flows, making the brand an asset with a certain value. This study primarily used the China Enterprise Brand Value Index published by the World Brand Lab to measure brand value. The World Brand Lab had strong expertise in brand value research, and its innovative evaluation method, the Brand Added-Value Assessment Model, has been widely recognized by the management and academic communities. The China's Top 500 Most Valuable Brands lists, published by the World Brand Lab, reflected the value of brands in the competitive environment of China and had significant influence both in China and globally [37–40]. In terms of data processing, as the sample data exhibited a skewed distribution, a logarithmic transformation was applied.

#### 3.2.2. Explanatory Variables

Corporate Social Responsibility (CSR): CSR was used as an explanatory variable in this study. The measurement of CSR was based on the ratings provided by third-party organizations. Currently, there are two mainstream organizations in China that provide CSR ratings, namely Hexun and RKS. However, due to the relatively low overall CSR ratings given by Hexun in 2018 and 2019, as well as the widespread absence of environmental responsibility scores, this study adopted the approach used by D. Zheng to measure CSR [41]. It quantified CSR by taking the average of the ratings from both organizations.

The use of the average value also incorporated the evaluations from both organizations, ensuring a more objective and fair assessment.

### 3.2.3. Moderating Variables

The study divided social capital into horizontal social capital (HC) and vertical social capital (VC). Based on existing research, horizontal social capital referred to the connections between companies and investors, suppliers, retailers, and customers. Horizontal social capital, as referenced by HC, mainly used the proportion of company executives holding concurrent positions in other companies. Vertical social capital referred to the social connections between companies and government agencies and officials. Vertical social capital (VC) primarily used the proportion of executives with government work experience, including whether they had previously worked in government departments, whether they were members of the National People's Congress, and whether they were members of the Chinese People's Political Consultative Conference [25–28]. (The executives of outstanding listed companies had the opportunity to be selected as members of the National People's Congress (NPC) and the Chinese People's Political Consultative Conference (CPPCC), which would increase their chances of obtaining government resources).

### 3.2.4. Control Variables

Previous research has shown that firm size (SIZE), ownership nature (STATE), managerial ability (TST), debt-paying ability (ALR), market competition (MC), advertising intensity (AD), years since listing (Listage), management ownership ratio (Fratio), and fixed asset ratio (Cratio) all had an impact on the relationship between corporate social responsibility and financial performance as well as brand value [42–45]. Therefore, this study selected the aforementioned variables as control variables. The specific variable descriptions in this study are as Table 1 [46–50].

**Table 1.** Variable definition.

| Variable Classification | Variable | Variable Symbol | Variable Description |
|---|---|---|---|
| Dependent variable | Corporate financial performance (short-term performance) | ROA | Return on total assets |
| | Brand value (long-term performance) | BD | Chinese Enterprise Brand Value Index |
| Explanatory variable | Corporate Social Responsibility | CSR | Annual CSR value of listed companies |
| Adjusting variables | Horizontal social capital | HC | The proportion of company executives working part-time in other enterprises |
| | Vertical social capital | VC | Proportion of executives with government work experience |
| Control variable | Enterprise size | SIZE | The ending balance of total assets is taken as 1 if it is greater than the median, and 0 if it is less than the median |
| | Nature of ownership | SOE | 1 for state-owned enterprises and 0 for non-state-owned enterprises (private or foreign-funded) |
| | Enterprise operational capability | TST | Total asset turnover rate = operating income/total asset balance at the end of the period |

**Table 1.** *Cont.*

| Variable Classification | Variable | Variable Symbol | Variable Description |
|---|---|---|---|
| Control variable | Financial leverage | LEV | Asset liability ratio = total liabilities/total assets |
| | Market competition level | MC | Degree of competition = sales expenses/revenue expenses |
| | Advertising intensity | AD | Advertising intensity = sales Management expenses/operating income |
| | Years of listing of enterprises | Listage | Year since the company was listed |
| | Management shareholding ratio | Fratio | Management shareholding ratio = Number of management shareholding/Number of A-shares issued by listed companies |
| | Fixed asset ratio | Cratio | Fixed asset ratio = total fixed assets/total assets |
| | Dummy variable | YEAR | Year |
| | Dummy variable | INDUSTRY | Industry |

*3.3. Model Settings*

To analyze the impact of corporate social responsibility on financial performance and brand value, this study constructed the following linear regression model:

$$ROA_{it} = \beta 0 + \beta 1 CSR_{it} + \beta 2 Controls_{it} + Year + Industry + \varepsilon_{it} \tag{1}$$

$$BD_{it} = \beta 0 + \beta 1 CSR_{it} + \beta 2 Controls_{it} + Year + Industry + \varepsilon_{it} \tag{2}$$

$$ROA_{it} = \beta 0 + \beta 1 CSR_{it} + \beta 2 HC_{it} + \beta 3 CSR_{it} \times HC_{it} + \beta 1 CSR_{it} \\ + \beta 3 Controls_{it} + Year + Industry + \varepsilon_{it} \tag{3}$$

$$ROA_{it} = \beta 0 + \beta 1 CSR_{it} + \beta 2 VC_{it} + \beta 3 CSR_{it} \times VC_{it} + \beta 1 CSR_{it} \\ + \beta 3 Controls_{it} + Year + Industry + \varepsilon_{it} \tag{4}$$

$$BD_{it} = \beta 0 + \beta 1 CSR_{it} + \beta 2 HC_{it} + \beta 3 CSR_{it} \times HC_{it} + \beta 1 CSR_{it} + \beta 3 Controls_{it} + \\ Year + Industry + \varepsilon_{it} \tag{5}$$

$$BD_{it} = \beta 0 + \beta 1 CSR_{it} + \beta 2 VC_{it} + \beta 3 CSR_{it} \times VC_{it} + \beta 1 CSR_{it} + \beta 3 Controls_{it} \\ + Year + Industry + \varepsilon_{it} \tag{6}$$

where $ROA_{it}$, $BD_{it}$, $CSR_{it}$, $HC_{it}$, and $VC_{it}$ represent the financial performance, brand value, corporate social responsibility, horizontal social capital, and vertical social capital of company *i* in year *t*, respectively; $Controls_{it}$ represents a series of control variables while Year and Industry represent the year and industry dummy variables, respectively; and $\varepsilon_{it}$ represents the error term.

## 4. Analysis of Empirical Results

*4.1. Descriptive Statistics and Correlation Analysis*

4.1.1. Descriptive Statistical Features

Table 2 presents the descriptive statistics of the variables, including the minimum, maximum, mean, and standard deviation. It could be observed that CSR had a large standard deviation, and the maximum value of the CSR level among the 81 companies was approximately 82.633, while the minimum value was around 22.201, indicating a significant difference in CSR levels. The original data for the brand value (BD) was skewed, but after performing the logarithm, the minimum value was approximately 0.782, the maximum

value was around 3.462, and the standard deviation was 0.583. The transformed sample followed a normal distribution. The mean value of the horizontal social capital was 0.621, indicating that most executives had experience working in other companies, indicating close connections with other enterprises. The mean value of the vertical connections (VC) was 0.244, suggesting that most executives in companies had little experience working in government departments, indicating fewer connections with the government. Additional explanations for other indicators can be found in Table 2.

**Table 2.** Descriptive statistics.

| Variable | Minimum Value | Maximum Value | Mean Value | Standard Deviation |
|---|---|---|---|---|
| CSR | 22.201 | 82.633 | 49.321 | 13.954 |
| ROA | 0.041 | 0.225 | 0.141 | 0.084 |
| BD | 0.783 | 3.462 | 2.182 | 0.583 |
| HC | 0.000 | 1.000 | 0.621 | 0.222 |
| VC | 0.000 | 1.000 | 0.244 | 0.131 |
| SIZE | 0.000 | 1.000 | 0.421 | 0.521 |
| SOE | 0.000 | 1.000 | 0.351 | 0.183 |
| TST | 0.222 | 0.812 | 0.450 | 0.312 |
| LEV | 0.061 | 0.913 | 0.511 | 0.172 |
| MC | 0.069 | 0.481 | 0.315 | 0.154 |
| AD | 0.051 | 0.461 | 0.189 | 0.091 |
| Listage | 2.000 | 28.000 | 13.850 | 6.670 |
| Fratio | 0.000 | 0.221 | 0.121 | 0.084 |
| Cratio | 0.215 | 0.681 | 0.412 | 0.221 |

### 4.1.2. Correlation Coefficient

Table 3 presents the correlation coefficients from the correlation analysis. By observing the correlational data in the table, it could be noted that corporate social responsibility (CSR) was significantly positively correlated with both return-on-assets (ROA) and brand value (BD) at a significance level of 5%, providing preliminary support for Hypotheses 1 and 2. Among the control variables, variables such as firm size, ownership nature, operational capability, financial leverage, market competitiveness, advertising intensity, and years since listing were all significantly positively correlated with both ROA and BD at a significance level of at least 10%, indicating that these control variables had some impact on firm financial performance. The specific effects of these variables need to be further examined in subsequent research.

**Table 3.** Correlation test.

| Variable | CSR | ROA | BD | HC | VC | SIZE | SOE | TST | LEV | MC | AD | Listage | Fratio | Cratio |
|---|---|---|---|---|---|---|---|---|---|---|---|---|---|---|
| CSR | 1 | | | | | | | | | | | | | |
| ROA | 0.167 *** | 1 | | | | | | | | | | | | |
| BD | 0.075 ** | 0.058 ** | 1 | | | | | | | | | | | |
| HC | 0.411 | 0.082 * | 0.013 * | 1 | | | | | | | | | | |
| VC | 0.011 | 0.257 * | 0.052 ** | 0.159 ** | 1 | | | | | | | | | |
| SIZE | 0.362 | 0.375 ** | 0.208 * | 0.322 * | 0.349 | 1 | | | | | | | | |
| SOE | 0.359 | 0.096 ** | 0.325 * | 0.013 * | 0.144 | 0.078 | 1 | | | | | | | |
| TST | 0.425 | 0.152 * | 0.301 * | 0.107 | 0.351 * | 0.104 | 0.164 | 1 | | | | | | |
| LEV | 0.208 | −0.202 ** | −0.325 * | 0.049 | 0.267 | 0.213 * | 0.161 | 0.318 | 1 | | | | | |
| MC | 0.379 | −0.059 ** | −0.011 ** | 0.264 | 0.161 | 0.175 * | 0.281 | 0.221 | 0.403 | 1 | | | | |
| AD | 0.066 | 0.279 * | 0.052 * | 0.301 | 0.215 * | 0.394 * | 0.155 | 0.421 * | 0.081 * | 0.048 | 1 | | | |
| Listage | 0.348 | 0.118 ** | 0.344 * | 0.169 * | 0.109 | 0.152 | 0.036 * | 0.37 * | 0.049 | 0.122 | 0.286 | 1 | | |
| Fratio | 0.052 | 0.079 ** | 0.283 | 0.051 ** | 0.356 * | 0.219 | 0.152 * | 0.271 * | 0.376 | 0.011 | 0.321 | 0.411 | 1 | |
| Cratio | 0.312 | 0.191 ** | 0.369 | 0.429 | 0.289 | 0.229 ** | 0.402 ** | 0.229 | 0.061 * | 0.174 | 0.199 | 0.195 | 0.081 | 1 |

Note: ***, **, and * represent the significance levels of regression coefficients at 1%, 5%, and 10%.

### 4.1.3. Collinearity Test

Before conducting the formal regression analysis, we performed a collinearity test to identify any potential collinearity issues among the variables. Collinearity problems between variables could lead to model distortion and result in incorrect estimations. Therefore, it was important to conduct a collinearity test. The results of the collinearity test are presented in Table 4, showing that the maximum variance inflation factor (VIF value) among the variables was 1.95. According to the general rule of thumb, when the VIF value was less than 10, we considered that there was no severe collinearity problem in the model, and we could proceed with the next step of the regression analysis.

**Table 4.** Collinearity test.

| Variable | ROA | BD | CSR | HC | VC | SIZE | SOE | TST | LEV | MC | AD | Listage | Fratio | Cratio |
|---|---|---|---|---|---|---|---|---|---|---|---|---|---|---|
| VIF value | 1.95 | 1.73 | 0.81 | 0.97 | 0.80 | 1.19 | 1.10 | 1.58 | 0.26 | 0.94 | 0.97 | 1.07 | 0.79 | 1.79 |

### 4.2. Empirical Research Results

#### 4.2.1. Benchmark Regression Results

The Hausman test was conducted to determine whether the panel data were suitable for a fixed-effect or random-effect model. The null hypothesis, H0: μi is not correlated with any explanatory variables, was rejected for all models (1–6), indicating that a fixed-effect model should be used. In the fixed-effect regression analysis of models (1–6), as shown in Table 5, (1–2) are the regression results for corporate social responsibility and brand value, (3–4) are the moderation-effect regression results for horizontal social capital, and (5–6) are the moderation-effect regression results for vertical social capital.

In regression (1), the estimated coefficient of corporate social responsibility (CSR) was 9.0322, and it was statistically significant at the 1% level. This suggested that CSR promoted firm performance (ROA), supporting Hypothesis 1. This was consistent with existing research [51].

In regression (2), the estimated coefficient of CSR was 7.1904, and it was statistically significant at the 1% level. This indicated that CSR positively influenced brand value (BD), supporting Hypothesis 2. This was consistent with existing research [52].

In regression (3), the estimated coefficient of CSR was 6.7010, and the estimated coefficient of CSR and horizontal social capital (HC) was 0.0832. Both coefficients were statistically significant at the 10% level. This suggested that HC positively moderated the relationship between CSR and firm performance (ROA), supporting Hypothesis 3a.

In regression (4), the estimated coefficient of CSR was 5.8249, and the estimated coefficient of CSR and HC was 0.0546. Both coefficients were statistically significant at the 5% level. This indicated that HC positively moderated the relationship between CSR and brand value (BD), supporting Hypothesis 3b.

In regression (5), the estimated coefficient of CSR was 6.6567, and the estimated coefficient of CSR and HC interaction term was 0.0112. Both coefficients were statistically significant at the 1% level. This suggested that HC positively moderated the relationship between CSR and firm performance (ROA), supporting Hypothesis 4a.

In regression (6), the estimated coefficient of CSR was 4.7664, and the estimated coefficient of CSR and HC interaction term was 0.0212. Both coefficients were statistically significant at the 1% level. This indicated that HC positively moderated the relationship between CSR and firm performance (ROA), supporting Hypothesis 4b.

Regarding the control variables, advertising intensity had a significant positive impact on both financial performance and brand value. Additionally, the leverage ratio was negatively correlated with financial performance at a level below 10% and negatively correlated with brand value at a 10% level. These findings aligned with real-life situations and indirectly validated the effectiveness of the data and model.

**Table 5.** Regression results.

| Variable | 1 | 2 | 3 | 4 | 5 | 6 |
|---|---|---|---|---|---|---|
| | ROA | BD | ROA | BD | ROA | BD |
| CSR | 9.0322 *** | 7.1904 *** | 6.7010 *** | 5.8249 *** | 6.6567 *** | 4.7664 *** |
| | (2.3539) | (0.0351) | (1.3516) | (1.3522) | (0.3527) | (0.3520) |
| HC | | | 0.4152 | 0.1342 | | |
| | | | (0.8149) | (0.5151) | | |
| HC × CSR | | | 0.0832 ** | 0.0546 ** | | |
| | | | (0.0415) | (0.2713) | | |
| VC | | | | | 0.0615 | 0.0394 |
| | | | | | (0.1495) | (0.8145) |
| VC × CSR | | | | | 0.0112 | 0.0212 |
| | | | | | (0.0004) | (0.0003) |
| SIZE | 0.0361 ** | 0.0031 * | 0.0362 * | 0.0291 * | 0.0361 | 0.0039 |
| | (0.0171) | (0.002) | (0.0212) | (0.0165) | (0.0484) | (0.0541) |
| SOE | 0.1650 | 0.2132 | 0.1750 | 0.3240 | 0.1662 | 0.9361 |
| | (0.4191) | (0.4721) | (0.2190) | (0.8820) | (0.5891) | (0.8832) |
| TST | 0.0982 | 0.0083 | 0.0778 | 0.0090 | 0.0798 | 0.0893 |
| | (0.1302) | (0.1101) | (0.1300) | (0.1210) | (0.1301) | (0.1310) |
| LEV | 1.2644 *** | 0.0349 *** | 1.5086 *** | 1.5258 *** | 1.5057 *** | 1.5263 *** |
| | (0.1221) | (0.0018) | (0.1279) | (0.1283) | (0.1280) | (0.1283) |
| MC | 0.0085 | 0.0079 | 0.0859 | 0.0063 | 0.0869 | 0.0693 |
| | (0.0231) | (0.1211) | (0.0931) | (0.1102) | (0.0331) | (0.1101) |
| AD | 0.1187 *** | 0.0050 *** | 0.0838 *** | 0.0854 *** | 0.0834 *** | 0.0850 *** |
| | (0.0212) | (0.0004) | (0.0211) | (0.0204) | (0.0203) | (0.0217) |
| Listage | 0.0148 | 0.0139 | 0.0209 | 0.0227 | 0.0309 | 0.0337 |
| | (0.1190) | (0.1290) | (0.2191) | (0.2291) | (0.3191) | (0.1854) |
| Fratio | 0.0082 | 0.0051 | 0.0847 | 0.0457 | 0.0891 | 0.0878 |
| | (0.0219) | (0.1132) | (0.0209) | (0.1130) | (0.0309) | (0.0113) |
| Cratio | 0.0097 | 0.0084 | 0.0974 | 0.0849 | 0.0376 | 0.0869 |
| | (0.0189) | (0.0199) | (0.0289) | (0.0599) | (0.0489) | (0.0999) |
| _cons | 2.2938 ** | 0.1429 *** | (0.0289) | (0.0599) | (0.0489) | (0.0999) |
| | (0.9367) | (0.0140) | (0.9461) | (1.1942) | (0.9544) | (1.2158) |
| Industry | Yes | Yes | Yes | Yes | Yes | Yes |
| Year | Yes | Yes | Yes | Yes | Yes | Yes |
| R2 | 0.3900 | 0.0433 | 0.3372 | 0.3333 | 0.3363 | 0.3296 |
| N | 810 | 810 | 810 | 810 | 810 | 810 |
| Hausman test | Prob > chi2 =0.0000 | Prob > chi2 =0.0000 | Prob > chi2 =0.0000 | Prob > chi2 =0.0000 | Prob > chi2 =0.0000 | Prob > chi2 =0.0000 |

Note: ***, **, and * represent the significance levels of regression coefficients at 1%, 5%, and 10%, respectively, with robust standard errors in parentheses.

### 4.2.2. Robustness Test Results
Winsorize

In this study, a variable replacement method was used to conduct a robustness test. The ROA variable was truncated at both ends using a 5% standard. The regression results are shown in Table 6.

After the truncation process, the coefficient estimates for CSR, CSRHC, and CSRVC remained consistent in terms of sign and significance with the previous analysis, indicating that the assumptions made in this study were robust.

Replacing the Dependent Variable

The dependent variable was subjected to robustness testing to determine the robustness of the previous conclusions. For short-term performance, we replaced the proxy variable, i.e., the return-on-assets (ROA), with the return-on-equity (ROE). For long-term performance, we used the annual Best China Brands ranking data published by Interbrand, instead of the previous brand value. The regression results are shown in Table 7, and it

could be observed that there was no significant difference between the various results and the previous ones, indicating the robustness of the results.

**Table 6.** Robustness test results.

| Variable | 1 | 2 | 3 | 4 | 5 | 6 |
| --- | --- | --- | --- | --- | --- | --- |
| | **ROA** | **BD** | **ROA** | **BD** | **ROA** | **BD** |
| CSR | 7.4228 *** | 5.2462 *** | 6.5362 *** | 4.6842 *** | 4.5005 *** | 3.6014 *** |
| | (2.3360) | (0.0360) | (2.3492) | (1.3483) | (0.3505) | (0.3486) |
| HC | | | 0.0062 | 0.0042 | | |
| | | | (0.0179) | (0.0169) | | |
| HC × CSR | | | 0.0341 *** | 0.0422 *** | | |
| | | | (0.0053) | (0.0098) | | |
| VC | | | | | 0.0036 | 0.0113 |
| | | | | | (0.0168) | (0.0229) |
| VC × CSR | | | | | 0.0332 *** | 0.0164 *** |
| | | | | | (0.0058) | (0.0016) |
| Controls | Yes | Yes | Yes | Yes | Yes | Yes |
| Industry | Yes | Yes | Yes | Yes | Yes | Yes |
| Year | Yes | Yes | Yes | Yes | Yes | Yes |
| R2 | 0.3819 | 0.0268 | 0.3595 | 0.3516 | 0.3585 | 0.3465 |
| N | 769 | 769 | 769 | 769 | 769 | 769 |

Note: *** represent the significance levels of regression coefficients at 1% with robust standard errors in parentheses.

**Table 7.** Robustness test results.

| Variable | 1 | 2 | 3 | 4 | 5 | 6 |
| --- | --- | --- | --- | --- | --- | --- |
| | **ROA** | **BD** | **ROA** | **BD** | **ROA** | **BD** |
| CSR | 7.0004 *** | 5.1112 *** | 6.5422 *** | 5.1640 *** | 4.2131 *** | 3.9344 *** |
| | (2.0010) | (0.0310) | (2.1291) | (1.124) | (0.0120) | (0.1453) |
| HC | | | 0.0098 | 0.0052 | | |
| | | | (0.0089) | (0.0039) | | |
| HC × CSR | | | 0.0454 *** | 0.0411 *** | | |
| | | | (0.0042) | (0.0041) | | |
| VC | | | | | 0.0074 | 0.0196 |
| | | | | | (0.0968) | (0.0779) |
| VC × CSR | | | | | 0.0452 *** | 0.0244 *** |
| | | | | | (0.0018) | (0.0092) |
| Controls | Yes | Yes | Yes | Yes | Yes | Yes |
| Industry | Yes | Yes | Yes | Yes | Yes | Yes |
| Year | Yes | Yes | Yes | Yes | Yes | Yes |
| R2 | 0.3921 | 0.2681 | 0.3544 | 0.3519 | 0.3247 | 0.3545 |
| N | 810 | 810 | 810 | 810 | 810 | 810 |

Note: *** represent the significance levels of regression coefficients at 1% with robust standard errors in parentheses.

Changing the Regression Method

To ensure the robustness of the results in this study, we conducted further robustness tests by employing the system of simultaneous-equations regression method. Specifically, we simultaneously combined models (1) and (2); models (3) and (4); and models (5) and (6). The regression analysis was then performed on these combinations, and the regression results are shown in Table 8. It could be observed that the regression results were largely consistent with those presented earlier, thus reaffirming the robustness of the conclusions drawn in this study.

**Table 8.** Robustness test results.

| Variable | 1 ROA | 2 BD | 3 ROA | 4 BD | 5 ROA | 6 BD |
|---|---|---|---|---|---|---|
| CSR | 5.0124 *** (1.0240) | 4.1219 *** (0.2390) | 5.5214 *** (2.0111) | 5.0121 *** (1.1041) | 4.9211 *** (0.0195) | 3.8345 *** (0.12257) |
| HC | | | 0.0211 (0.0212) | 0.0124 (0.0039) | | |
| HC × CSR | | | 0.0777 *** (0.0072) | 0.0787 *** (0.0039) | | |
| VC | | | | | 0.0757 (0.0815) | 0.0978 (0.0975) |
| VC × CSR | | | | | 0.0554 *** (0.0028) | 0.0317 *** (0.0054) |
| Controls | Yes | Yes | Yes | Yes | Yes | Yes |
| Industry | Yes | Yes | Yes | Yes | Yes | Yes |
| Year | Yes | Yes | Yes | Yes | Yes | Yes |
| R2 | 0.3012 | 0.2015 | 0.3111 | 0.3245 | 0.3547 | 0.3547 |
| N | 810 | 810 | 810 | 810 | 810 | 810 |

Note: *** represent the significance levels of regression coefficients at 1% with robust standard errors in parentheses.

### 4.2.3. Discussion on Endogeneity Issues

This study needed to consider two aspects of endogeneity. Firstly, there may have been a reverse causality, where companies with higher performance and brand value may have had stronger political backgrounds and economic strength, making it easier for them to fulfill their social responsibilities. Therefore, this study regressed the core explanatory variable one period in advance to weaken the impact of reverse causality. Regression (1–2) in Table 9 shows that the coefficient of the lagging corporate social responsibility (CSR) on firm performance and brand value had the same direction and was significant, consistent with the original model.

**Table 9.** Early explanatory variables and IV-2SLS regression results.

| Variable | 1 ROA | 2 BD | 3 CSR | 4 ROA | 5 CSR | 6 BD |
|---|---|---|---|---|---|---|
| CSR-1 | 5.4018 *** (0.3029) | 6.1677 *** (0.0357) | | | | |
| religion | | | 4.4931 *** (1.1707) | | 5.4544 *** (0.9541) | |
| IV | | | | 5.1438 *** (1.4817) | | 7.1938 *** (1.4817) |
| Minimum eigenvalue statistic | | | | 23.1140 | | 33.2350 |
| Controls | Yes | Yes | Yes | Yes | Yes | Yes |
| Industry | Yes | Yes | Yes | Yes | Yes | Yes |
| Year | Yes | Yes | Yes | Yes | Yes | Yes |
| R2 | 0.2498 | 0.0241 | 0.0251 | 0.3238 | 0.2948 | 0.3448 |
| N | 810 | 810 | 810 | 810 | 810 | 810 |

Note: *** represent the significance levels of regression coefficients at 1% with robust standard errors in parentheses.

Secondly, this study may have omitted variables. Therefore, an instrumental variable (IV) analysis was used to address endogeneity. Following the approach of D. Chen [53], the number of religious activity places in the company's location registration was selected as the instrumental variable for the Two-Stage least squares (2SLS) regression. D. Chen's research found a positive relationship between executives' religious beliefs and corporate social responsibility [54]. Therefore, in areas with more religious activity places, there could have been more people with religious beliefs and a stronger religious atmosphere [55]. This may have affected the awareness of owners, senior management, and employees regarding fulfilling their own social responsibility, thereby influencing the behaviors influencing corporate social responsibility. Additionally, the number of religious activity places in

a company's location registration was unrelated to other control variables and random disturbances. Therefore, using the number of religious activity places in the company's location registration as an instrumental variable for the tone of the corporate social responsibility reporting was reasonable. The data on the number of religious activity places in the company's location registration were obtained from the National Religious Affairs Bureau's Religious Basic Information Query System.

The results of the IV-2SLS regression are shown in Table 7 (3–6). In the first-stage regression, the coefficient of the instrumental variable (IV) was 7.1238 and significant at the 1% level, indicating a strong correlation between the instrumental variable and the explanatory variable. According to the Stock–Yogo test, the minimum eigenvalue statistics of 23.114 and 23.325 were both greater than 10% of the critical value of 19.93, further indicating that this study did not suffer from weak instrument problems. The instrumental variable estimation results showed that the regression coefficients of corporate social responsibility (CSR) on firm performance and brand value were 5.1438 and 7.1938, respectively, and both were significant at the 1% level, further validating the research hypothesis of this study.

4.2.4. Extensibility Research
Group Testing of Compliant Disclosure and Voluntary Disclosure

This study focused on listed companies that disclosed corporate social responsibility (CSR) reports. Among these companies, some were required to disclose their CSR reports, while others did so voluntarily. (The Shanghai and Shenzhen Stock Exchanges stipulated that sample companies in the corporate governance sector, companies issuing overseas-listed foreign shares, and financial companies had to disclose social responsibility reports, abbreviated as compliant disclosure). By comparing the impact of CSR performance on financial performance between the two groups, more targeted corporate governance recommendations could be proposed. Among the 81 samples selected in this study, 49 were mandatory-disclosure companies and 32 were voluntary-disclosure companies. The average CSR score for the 49 mandatory-disclosure companies was 47.68, while the average score for the 32 voluntary-disclosure companies was 44.58. The average scores for both types of companies were not high, and the difference between the average scores of the mandatory and voluntary-disclosure companies was only three points. This indicated that although mandatory disclosure had been required, regulatory authorities in China did not have strict requirements for disclosure content, resulting in a relatively small difference in average scores between mandatory and voluntary disclosure. There was still room for improvement. In order to study whether there was a significant difference concerning the impact of voluntary or mandatory disclosure on firm performance, this study conducted empirical research on all samples, categorized by disclosure type.

The results of the grouped regression are shown in Table 10. Columns (1–6) show the regression results for mandatory-disclosure companies, and columns (7–12) show the regression results for voluntary-disclosure companies. The adjusted R2 values for both groups were good, indicating a good fit of the model. The results showed that both mandatory- and voluntary-disclosure companies had a positive correlation between CSR and firm performance as well as brand value, significant at the 1% level. From the perspective of correlation coefficients, in the mandatory-disclosure group, the effect of the corporate social value on the corporate financial performance was greater than the effect on brand value, while in the voluntary-disclosure group, the effect of the corporate social responsibility on the brand value was greater than the effect on the corporate financial performance. This suggested that companies with a strong sense of social responsibility were more likely to gain public trust and, thus, achieve long-term brand value. Finally, based on the results of the moderation-effect test, it was found that both the interaction term of horizontal social resources (HC) and corporate social responsibility (CSR), and the interaction term of vertical social resources (VC) and corporate social responsibility (CSR) were significant, at minimum, at the 5% level, indicating that horizontal social resources

(HC) and vertical social resources (VC) continued to play a moderating role in different groups. Additionally, based on the regression coefficients, in the voluntary-disclosure group, the moderation-effect coefficients of vertical social capital were 0.0745 and 0.0638, which were greater than the moderation-effect values of horizontal social capital, 0.0266 and 0.0264. In the mandatory-disclosure group, there was no significant difference in the moderation effect of horizontal social capital and vertical social capital, indicating that companies that had voluntarily disclosed their social responsibility had been more likely to obtain government resources due to their easier fulfillment of corporate social responsibility, leading to improved corporate performance and brand value.

**Table 10.** Compliance disclosure and voluntary disclosure of regression results.

| Variable | Regulatory Disclosure | | | | | | Voluntary Disclosure | | | | | |
|---|---|---|---|---|---|---|---|---|---|---|---|---|
| | 1 ROA | 2 BD | 3 ROA | 4 BD | 5 ROA | 6 BD | 7 ROA | 8 BD | 9 ROA | 10 BD | 11 ROA | 12 BD |
| CSR | 8.1481 *** (1.0100) | 4.8068 *** (0.1850) | 6.3355 ** (0.156) | 4.0294 *** (0.0029) | 7.3355 ** (0.1726) | 5.0294 *** (0.1419) | 5.2935 *** (0.3932) | 7.1387 *** (0.0326) | 4.3235 ** (0.1121) | 8.0294 *** (0.0179) | 4.3715 ** (0.1591) | 6.0294 *** (0.0759) |
| HC | | | 0.3355 (0.1564) | 0.0294 (0.0529) | | | | | 0.2366 (0.2664) | 0.0214 (0.0629) | | |
| HC × CSR | | | 0.0255 ** (0.0121) | 0.0341 *** (0.0089) | | | | | 0.0266 ** (0.1301) | 0.0264 *** (0.0084) | | |
| VC | | | | | 0.0345 (0.165) | 0.0214 (0.0319) | | | | | 0.0766 (0.0064) | 0.0894 (0.0019) |
| VC × CSR | | | | | 0.0215 *** (0.00176) | 0.0324 *** (0.0029) | | | | | 0.0745 *** (0.0041) | 0.0638 *** (0.0089) |
| Controls | Yes | Yes | Yes | Yes | Yes | Yes | Yes | Yes | Yes | Yes | Yes | Yes |
| Industry | Yes | Yes | Yes | Yes | Yes | Yes | Yes | Yes | Yes | Yes | Yes | Yes |
| Year | Yes | Yes | Yes | Yes | Yes | Yes | Yes | Yes | Yes | Yes | Yes | Yes |
| R2 | 0.3240 | 0.2981 | 0.2921 | 0.2650 | 0.3250 | 0.3010 | 0.2731 | 0.2830 | 0.3240 | 0.3080 | 0.3291 | 0.2770 |
| N | 490 | 490 | 490 | 490 | 490 | 490 | 320 | 320 | 320 | 320 | 320 | 320 |

Note: ***, **, represent the significance levels of regression coefficients at 1% and 5% respectively, with robust standard errors in parentheses.

Group Testing of Heavy-Polluting and Non-Heavily-Polluting Enterprises

In recent years, the country has attached great importance to environmental protection. Against the backdrop of carbon neutrality and carbon targets included in government work reports, the green economy has become a trend in human development, and thus the role played by heavy-pollution enterprises was particularly important. According to the relevant regulations of the Ministry of Environmental Protection, China had designated 16 industries as heavy-pollution industries. (According to the Guidelines for Environmental Information Disclosure of Listed Companies, published by the Ministry of Environmental Protection, heavy-pollution industries included 16 categories, such as thermal power, steel, cement, electrolytic aluminum, coal, metallurgy, chemical industry, petrochemical industry, building materials, papermaking, brewing, pharmaceuticals, fermentation, textiles, tanning, and mining). Therefore, this study categorized the samples as heavy-pollution industries and non-heavy-pollution industries for group testing, further studying the difference in the impact of CSR performance on financial performance in different industries. Among all the samples, there were a total of 21 companies in the heavy-pollution group and 60 companies in the non-heavy-pollution group. Among them, the average CSR score for heavy-pollution group was 35.85, slightly lower than the average score of 36.34 for the other group.

The regression results are shown in Table 11. The regression results showed that the impact of corporate social responsibility on the company's performance (ROA) and brand value (BD) was significant at the 1% level in both groups. For non-heavy-pollution companies, CSR had a positive effect on both firm performance and brand value, with little difference between the two. However, for heavy-pollution companies, the coefficient of CSR on brand value was higher than that on firm performance, indicating that heavy-pollution companies could gain greater brand value by fulfilling CSR.

**Table 11.** Regression results of heavy-polluting and non-heavy-polluting enterprises.

| Variable | Non-Heavy-Polluting Industries | | | | | | Heavy-Polluting Industries | | | | | |
|---|---|---|---|---|---|---|---|---|---|---|---|---|
| | 1 ROA | 2 BD | 3 ROA | 4 BD | 5 ROA | 6 BD | 7 ROA | 8 BD | 9 ROA | 10 BD | 11 ROA | 12 BD |
| CSR | 7.0470 *** (0.0101) | 6.7057 *** (0.0175) | 5.2255 *** (0.055) | 6.0294 *** (0.0029) | 6.2255 ** (0.0725) | 5.0294 *** (0.0409) | 5.2925 *** (0.2922) | 7.0277 *** (0.0225) | 4.2225 ** (0.0020) | 7.0294 *** (0.0079) | 4.2705 ** (0.0590) | 5.0294 *** (0.0759) |
| HC | | | 0.2354 (0.0554) | 0.0273 (0.0529) | | | | | 0.2365 (0.2554) | 0.0294 (0.0548) | | |
| HC × SCR | | | 0.0255 ** (0.054) | 0.0154 *** (0.0079) | | | | | 0.0455 ** (0.0200) | 0.0554 *** (0.0069) | | |
| VC | | | | | 0.0245 (0.0555) | 0.0204 (0.0209) | | | | | 0.0355 (0.0054) | 0.0494 (0.0009) |
| VC × CSR | | | | | 0.0505 *** (0.0008) | 0.0624 *** (0.0029) | | | | | 0.0245 *** (0.0040) | 0.0227 *** (0.0079) |
| Controls | Yes | Yes | Yes | Yes | Yes | Yes | Yes | Yes | Yes | Yes | Yes | Yes |
| Industry | Yes | Yes | Yes | Yes | Yes | Yes | Yes | Yes | Yes | Yes | Yes | Yes |
| Year | Yes | Yes | Yes | Yes | Yes | Yes | Yes | Yes | Yes | Yes | Yes | Yes |
| R2 | 0.3140 | 0.2650 | 0.2500 | 0.2950 | 0.2770 | 0.2810 | 0.3150 | 0.3280 | 0.2840 | 0.2850 | 0.2800 | 0.3000 |
| N | 600 | 600 | 600 | 600 | 600 | 600 | 210 | 210 | 210 | 210 | 210 | 210 |

Note: ***, **, represent the significance levels of regression coefficients at 1% and 5% respectively, with robust standard errors in parentheses.

The moderating-effect study found that both horizontal social capital (HC) and vertical social capital (VC) had a positive moderating effect on the main regression. For non-heavy-pollution industries, the moderating effect of horizontal social capital (HC) was smaller than that of vertical social capital (VC). This was because there were a large number of private enterprises in non-heavy-pollution companies, and fulfilling more social responsibilities was a response to government policies in order to more easily obtain government support, thereby promoting the improvement of corporate performance and brand value. On the contrary, for the heavy-pollution industries, the horizontal social capital (HC) was greater than the vertical social capital (VC). This was mainly because heavy-pollution companies in China often referred to heavy-industry enterprises controlled by the government, such as thermal power, steel, cement, electrolytic aluminum, coal, metallurgy, etc. These enterprises were more likely to cause environmental problems in their production processes and attract public dissatisfaction [56]. However, their fulfillment of social responsibility, such as disclosing production pollution, was more likely to gain public support, thereby promoting firm performance and improving brand value.

## 5. Suggestions and Prospects

### 5.1. Conclusions

This study empirically analyzed the impact of CSR on firm performance and brand value, as well as the moderating effect of social capital. The results showed that CSR had a significant positive correlation with financial performance and brand value, in other words, the higher the level of CSR, the higher the firm's financial performance and brand value. Horizontal social capital played a moderating role in the impact of CSR on financial performance and brand value, that is, the higher the level of horizontal social capital, the more significant the positive impact of CSR on firm financial performance. Vertical social capital also moderated the impact of CSR on firm performance and brand value, that is, the higher the level of vertical social capital, the more significant the impact of CSR on increasing firm profits, gaining reputation, and increasing brand value.

The moderation-effect findings indicated that both the interaction term of horizontal social resources (HC) and corporate social responsibility (CSR), and the interaction term of vertical social resources (VC) and corporate social responsibility (CSR) were significant, at minimum, at the 5% level. Additionally, based on the grouped regression coefficients in the voluntary-disclosure group, the moderation-effect coefficients of vertical social capital were greater than the moderation-effect coefficients of horizontal social capital. In the mandatory-disclosure group, there was no significant difference in the moderation effects of horizontal social capital and vertical social capital. Furthermore, in the subgroup analysis

of heavy-pollution and non-heavy-pollution industries, it was found that both horizontal social capital (HC) and vertical social capital (VC) had a positive moderating effect on the main regression. For the non-heavy-pollution industry, the moderation-effect value of horizontal social capital (HC) was smaller than the moderation-effect value of vertical social capital (VC).

### 5.2. Discussion of Results

Current corporate social responsibility (CSR) has had a significant positive impact on financial performance, which was consistent with most previous research findings. This indicated that the impact of CSR on financial performance was immediate.

CSR also had a positive and significant impact on brand value, although the regression coefficients were generally smaller, as compared to the impact on firm performance. This was because the transmission of the CSR information was slower, and its underlying mechanism involved influencing consumer and supplier perceptions of the brand, improving brand satisfaction and loyalty, and, ultimately, affecting brand value. This lag effect determined the delayed impact of CSR on brand value.

The relationship between CSR, financial performance, and brand value was moderated by horizontal social capital. The impact of CSR on financial performance increased with higher levels of social capital. Previous research had suggested that higher levels of social capital enabled firms to obtain greater financial performance improvements through fulfilling social responsibility. Social capital strengthened collaborative relationships between firms and partners, enhanced communication, and increased the firm's ability to mitigate risks based on sharing management techniques, thereby providing greater positive financial performance effects due to CSR.

The relationship between CSR and financial performance as well as brand value was also moderated by vertical social capital. The impact of CSR on brand value increased at higher levels of vertical social capital. This study innovatively explored the moderating role of social capital between CSR and brand value, which had rarely been mentioned in previous research. Vertical social capital provided policy advantages, such as tax benefits and land policies, to firms and increased the likelihood of government protection during crises, which led to the recognition of the firm's strength by consumers and suppliers. According to signal theory, investors and consumers perceived that firms with lower levels of vertical social capital were at a competitive disadvantage and lacked strength. This perception may lead investors to question the firm's strategic planning, although it does not necessarily reduce brand value. The importance of the moderating role of vertical social capital implied that future research could further explore the other conditions and boundaries under which this relationship exists. The success of a brand's competitive strategy depended on identifying the points of differentiation from competitors.

### 5.3. Research Suggestions

Based on the research findings, this study proposed targeted recommendations from the perspectives of both enterprises and the government.

Enterprises should establish an accurate awareness of social responsibility and consider factors such as the environment, social development, and the interests of stakeholders in business activities. Enterprises should operate with a sustainable development mindset and not sacrifice environmental protection and the protection of consumer and stakeholder interests, solely for profitable pursuits. The importance of fulfilling social responsibility should be considered at the strategic level and should not be limited to the short-term benefits. For example, the automotive industry should promote energy conservation, emission reduction, and environmental-protection awareness. The construction and decoration industry should focus on addressing environmental pollution, noise pollution, dust pollution, etc. The biopharmaceutical industry should consider how to improve the accessibility of drugs for the general public. The research findings of this study also showed that better fulfillment of social responsibility by enterprises led to higher financial

performance. Therefore, establishing an accurate awareness of social responsibility is the first thing that Chinese enterprises should do, especially for heavily polluting industries. While these industries provide economic benefits to society, they should also prioritize sustainable social development. In the context of China's high attention to the ecological environment, heavy-polluting industries should take set an example by reducing environmental pollution through technological upgrades, better fulfilling their social responsibility, proactively disclosing CSR reports, and improving the quality of these reports. According to the research findings of this study, proactively disclosing CSR reports had a greater impact on brand value. Proactively disclosing CSR reports could also increase information transparency. Therefore, even for enterprises that are not required to publish CSR reports, they could improve brand value by proactively disclosing them.

Governments should be aware that creating a favorable social environment relies on the role they play. Currently, China's government has transitioned from an omnipotent government to a service-oriented government, allowing various market entities to compete freely. However, this does equal a hands-off approach. In the field of social responsibility, we need to start from the perspective of the masses and effectively utilize governmental supervision to enhance people's well-being. There have been difficulties in government supervision, primarily because China currently lacks legislation specifically targeting social responsibility. Therefore, this study provides direction for improving government supervision. In addition to mandatory measures, we also need to change people's perceptions by raising corporate social responsibility awareness and encouraging enterprises to publish social responsibility reports. Currently, the management of corporate social responsibility in China is not yet well-developed, so it is crucial to improve the awareness of fulfilling social responsibility at its root.

**Author Contributions:** Conceptualization, J.Z.; methodology, J.Z.; software, J.Z.; validation, J.Z.; formal analysis, J.Z.; investigation, J.Z.; resources, J.Z.; data curation, J.Z.; writing—original draft preparation, J.Z.; writing—review and editing, J.Z.; visualization, J.Z.; supervision, Z.L.; project administration, Z.L. All authors have read and agreed to the published version of the manuscript.

**Funding:** This research received no external funding.

**Institutional Review Board Statement:** Not applicable.

**Informed Consent Statement:** Informed consent was obtained from all subjects involved in the study.

**Data Availability Statement:** The data presented in this study are available in a public dataset.

**Conflicts of Interest:** The authors declare no conflict of interest.

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
