# Peer review of "The Impact of Corporate Social Responsibility on Financial Performance and Brand Value"

_sustainability, doi:10.3390/su152416864_

Round 1

Reviewer 1 Report

Comments and Suggestions for Authors

This paper assesses the impact of corporate social responsibility (CSR) on financial performance and brand value by using a dataset comprising of 81 Chinese listed companies. The results of the research point that CSR has a significant positive correlation with financial performance and brand value.

The topic of the paper is of interest for researchers and companies.

I find the paper to be well written and straightforward. The econometric work is well executed. The results and discussion are intuitive and comprehensive. The conclusions are sound and based on the empirical findings.

Perhaps one area in which improvements could be done is related to the literature review: more details on how variables were chosen and the results obtained by other researchers would be very useful.

Also, a more detailed comparison of the results from the paper with the ones from the related literature would enhance the value of the paper.

Author Response

Thank you very much for recognizing the work in this paper. Your professional feedback has greatly improved the quality of the paper. We carefully considered each of your suggestions and made corresponding revisions. The specific revisions and responses are as follows:

Feedback 1: Perhaps one area in which improvements could be done is related to the literature review: more details on how variables were chosen and the results obtained by other researchers would be very useful.

Response: In response to your feedback, we have added 22 references, focusing on the theoretical analysis, variable selection, and comparison of results with existing research.

Feedback 2: Also, a more detailed comparison of the results from the paper with the ones from the related literature would enhance the value of the paper.

Response: We completely agree with your suggestion and have therefore added references, especially for comparing empirical results with existing research.

To facilitate your review of the revised version, all our changes have been made in track changes mode. Once again, thank you for your professional feedback, and we wish you a pleasant day.

Reviewer 2 Report

Comments and Suggestions for Authors

The abstract of the article does not clearly explain the aim of the paper, nor does it mention specific benefits or results. I recommend moving away from general formulations and focusing more on precise results so that the abstract can better entice potential interested readers.
I recommend that the paper's title be simplified and made more concise.
The keywords are appropriately chosen, although they are general rather than explicitly defining the article.
The introduction of the article is not entirely typical of scientific publications, where the author, after a brief introduction of the concept, immediately moves on to regional examples. The next part of the text is more concerned with discussing a few ideas rather than covering the overall issue.
Chapter 2.2 is unclear, and although it contains several hypotheses, it is not at all clear on what assumptions and studies they are based on. Why do the authors address them at all? What good is it?
The text contains XXXX instead of specific terms.
The selection of enterprises is not appropriately defined; it is not clear why these particular enterprises.
The number of decimal places is not consistent.
Statistical outputs are correct, but why the authors address them is unclear.
Table 5 is unclear.
Overall, the paper relies on data from a small number of enterprises. In doing so, the authors do not adequately argue the need for the research the reasons for setting hypotheses. At the same time, the text lacks a comprehensive theoretical overview. The results are correctly processed, but the contribution of these results is not clearly explained, as well as their comparison with similar results. I do not recommend the article for publication, although it deals with an exciting topic, but methodologically, it is not properly conceived.

Author Response

Thank you very much for your recognition of the work in this article. Your professional feedback has greatly improved this article. We have carefully considered each of your review comments and made corresponding revisions. The specific modifications and responses are as follows:

Opinion 1. The abstract of the article does not clearly explain the aim of the paper, nor does it mention specific benefits or results I recommend moving away from general formulas and focusing more on precision results so that the abstract can be better entity potential interested readers

Reply: Based on your feedback, we have rewritten the abstract section, hoping to express it more concisely and fluently.

Opinion 2: I recommend that the paper's title is simplified and made more consensus

Reply: Based on your feedback, we have made adjustments to the title of the article, mainly by removing subheadings to make the title more concise.

Opinion 3: The keywords are appropriately chosen, although they are general rather than explicitly defining the article

Reply: We agree with your opinion and have added keywords, mainly focusing on the concept of social capital. In this article, it is one of the focuses, so we have included it in the keywords.

Opinion 4: The introduction of the article is not entirely typical of scientific publications, where the author, after a brief introduction of the concept, immediately moves on to regional examples The next part of the text is more concerned with discussing a few ideas rather than covering the overall issue

Reply: In the introduction section, our previous writing was not concise and fluent enough, so we have rewritten and revised the entire introduction section.

Opinion 5: Chapter 2.2 is unclear, and although it contains several hypotheses, it is not at all clear on what assessments and studies they are based on. Why do the authors address them at all? What good is it?

Reply: We have rewritten the question about the research hypothesis, reorganized the logical relationship of the research hypothesis, and added references to enhance persuasiveness.

Opinion 6: The text contains XXXX instead of specific terms

Reply: This incorrect expression has been corrected.

Opinion 7: The selection of enterprises is not appropriately defined; It is not clear why these particular enterprises

Reply: We agree with your opinion. In the revised version, we have modified and added the wording regarding the scope of selection for the help book, as well as why it was chosen in this way. One of our dependent variables is the brand value of the enterprise. Referring to existing research on China issues, most of them come from China's Top 500 Most Valuable Brands. Therefore, this article uses this list as the data source. Secondly, considering that the definition of brand value in this article is a long-term concept (brand value requires long-term accumulation), we choose enterprises that have been listed continuously for more than 10 years as our research sample.

Opinion 8: The number of decimal places is not consistent

Reply: Modified

Opinion 9: Statistical outputs are correct, but why the author address them is unclear

Reply: We have added a discussion on the ideas behind the empirical results of this article.

Opinion 10: Table 5 is unclear

Reply: We have added an explanation about Table 5.

In order to facilitate your review of the revised version, all our modifications are made in revision mode. Thank you again for your professional review comments. Wishing you a happy life.

Reviewer 3 Report

Comments and Suggestions for Authors

General remarks:

1. Stylistic merits of the paper need the improvement. Help of a competent native speaker is welcome.

2. I woud suggest to reference more more recent publications in the bibliography.

3. There are also some other editorial and language issues in the paper. Some of them are presented immediately below:

Page 5

XXXX

Page 6 

xxxxxxxxx

Page 7

The mean value of corporate social capital

-->

The mean value of horizontal corporate social capital

Page 14

The regression results after group testing are reported in the figure above.

---

IMHO it isn't clear which Figure is referenced here!

Page 16

the moderating effect of horizontal social capital (VC)

-->

... capital (HC)

Page 18

Increase awareness of social responsibility and encourage enterprises to publish social responsibility reports.

---

A misplaced sentence?

Comments on the Quality of English Language

There are some language issues in the paper. Language style has to be improved. 

Author Response

Thank you very much for your recognition of the work in this article. Your professional feedback has greatly improved this article. We have carefully considered each of your review comments and made corresponding revisions. The specific modifications and responses are as follows:

Opinion 1. Stylistic merits of the paper need the improvement Help of a competent native speaker is welcome

Reply: We have rechecked the language expression issues in the article and made extensive modifications based on this. Thank you for your feedback.

Opinion 2. I would suggest to refer more recent publications in the bibliography

Reply: We agree with your opinion and have added 22 references in the revised version to enhance the persuasiveness of this article. The increase in references mainly focuses on the research hypothesis section, variable selection section, and result explanation section.

Opinion 3. There are also some other editorial and language issues in the paper Some of them are presented immediately below:

Page 5

XXXX

Page 6

Xxxxxxxxxx

Page 7

The mean value of corporate social capital

The mean value of horizontal corporate social capital

Page 14

The regression results after group testing are reported in the figure above

IMMO it is not clear which Figure is referenced here!

Page 16

The modeling effect of horizontal social capital (VC)

... capital (HC)

Page 18

Increase awareness of social responsibility and resource enterprises to publish social responsibility reports

A missed presence?

Reply: We have reviewed each of the above questions and made revisions one by one. Thank you very much for your detailed feedback.

In order to facilitate your review of the revised version, all our modifications are made in revision mode. Thank you again for your professional review comments. Wishing you a happy life.

Reviewer 4 Report

Comments and Suggestions for Authors

See my attached file.

Comments on the Quality of English Language

The English writing of the manuscript is clear and generally well-structured. There are no significant grammatical errors, and the meaning is conveyed with clarity. However, there are minor issues related to formatting consistency, and a few sentences could be further refined for academic rigor. For example: The academic tone could be enhanced by avoiding colloquial expressions like "profit-first" mentality. The use of "will" in the last paragraph of the introduction to describe the paper's content should be in the present tense to maintain academic formality, e.g., "This paper explores" rather than "This paper will explore".

Author Response

Thank you very much for your recognition of the work in this article. Your professional feedback has greatly improved this article. We have carefully considered each of your review comments and made corresponding revisions. The specific modifications and responses are as follows:

The paper provides an empirical analysis of the relationship between CSR, financial performance, and brand value for Chinese listed companies The study is well structured To assist with possible further revisions, I would like to offer the following recommendations for the author's consideration:

Opinion 1. The research sample is limited to 81 companies, a relatively small fraction of the total companies listed on the China A-share market I resource the author to address the representativeness of this sample size within the study and to consider both the conclusions drawn are sufficiently generalizable The sample appearances to be more of a subgroup, and extrapolating findings from it to the broker population of listed companies may not be appropriate

Reply: We agree with your opinion. In the revised version, we have added information about the sample. In the revised version, we have modified and added wording about the selection range of the sample, as well as why it was chosen in this way. One of our dependent variables is the brand value of the enterprise. Referring to existing research on China issues, most of them come from China's Top 500 Most Valuable Brands. Therefore, this article uses this list as the data source. Secondly, considering that the definition of brand value in this article is a long-term concept (brand value requires long-term accumulation), we choose enterprises that have been listed continuously for more than 10 years as our research sample.

Opinion 2. Give the restrictive nature of the criteria to qualify for "China's Top 500 Most Valuable Brands," I recommend expanding the selection to avoid an over concentration on large mainstream industry companies This approach would limit potential sample selection bias and contribution to a more inclusive sample representation

Reply: We fully understand your concerns. Due to the impact of corporate social responsibility on brand value explored in this article, and the current research on the Chinese sample mostly chooses China's Top 500 Most Valuable Brands for brand value measurement, we believe that brand value and corporate financial performance are a relative concept in this article, one long-term and the other short-term (brand value requires time accumulation), Therefore, this article selects companies that have been on the list for 10 consecutive years. Of course, we acknowledge your opinions and concerns, so in the revised version, we have added explanations on this issue and added new references to assist with clarification.

Opinion 3. There is an inconsistency on page five regarding the source of CSR data, with section 3.1 citing the RKS database and section 3.2.2 specifically including Hexun in the calculation Additionally, a typographical error observes the reference for CSR performance measurement Clarification and correction of these points are necessary for the reader's understanding

Reply: Thank you for your careful review. We have added additional information about the database.

Opinion 4. The explanation of social capital, a key variable in this study, is not a brief, and the rational for referencing articles 27 and 28 is not clearly articulated A more comprehensive description of these variables and their empirical significance would enhance the reader's comprehension

Reply: As you said, our discussion on social capital is insufficient. In the revised version, we have added a discussion on social capital and added annotations and literature support to address this issue in the variable description section.

Opinion 5. There is a typographical error with 'xxxxxxx' in the explanation of control variables on page six

Reply: Corrected.

Opinion 6. The study's relationship on ROA as the sole measure of company performance is an oversimplification I suggest incorporating additional variables to conduct a more through robustness test

Reply: Thank you for your feedback. This article only conducted a robustness test, which is not enough to demonstrate the robustness of the results. Based on your suggestion, we have added ROE to replace short-term performance and replaced long-term performance according to the China Best Brand Ranking released by INTERBRAND. This is used for robustness testing, and the robustness results can be found in the revised section 4.2.2.2, Replace the dependent variable.

Opinion 7. I recommend using multiple equation regression for models (1) and (2) as a robustness check, with a similar approach for models (3) and (4), as well as (5) and (6)

Reply: In the robustness testing section, we have added a robustness test for replacing regression methods. Based on your suggestion, we will use a simultaneous equation model to regress (1) and (2) simultaneously, (3) and (4) simultaneously, and (5) and (6) simultaneously again. The results are shown in 4.2.2.3 Change the regression method.

Opinion 8. The paper's citations and references exhibit several non-standard practices A through review and correction of these sections are guaranteed

Reply: We have conducted inspections and modifications.

Opinion 9. The formatting inconsistencies in the regression analysis tables merit close

Examination and standardization

Reply: We have checked the paste errors in the regression results and made corrections.

  1. The sample size varies across different regression analyses need to be addressed and adjusted

Reply: We have checked the differences in the regression samples and provided explanations

Opinion 11. The sections titled 4.2.1 and 4.2.2 both describe Benchmark regression results without distinguishing between the two I recommend revising these sections to reflect their distinct contents or reconstructing them for clarity

Reply: We have made adjustments to this section by changing 4.2.2 to robustness testing.

Opinion 12. While the study focuses on Chinese companies, there is scan justification provided for the specific use of Chinese data in the research context An Elaboration on the necessity or motivation behind this choice would be beneficial

Reply: We fully agree with your opinion and have added an explanation in the introduction section on why China is used as a sample in this article. Added clarification on the source of data.

In addition, we have checked, revised, and adjusted the language of the article. Thank you for your professional feedback. In order to facilitate your review of the revised version, all our modifications are made in revision mode. Thank you again for your professional review comments. Wishing you a happy life.

Round 2

Reviewer 2 Report

Comments and Suggestions for Authors

The authors have taken all my comments into account, for this reason I have no further comments on the article. 

Author Response

Thank you for your recognition of our work, and we are especially grateful for your professional and detailed review comments. Your feedback has greatly contributed to the improvement of our article. We sincerely wish you a pleasant life.

Reviewer 4 Report

Comments and Suggestions for Authors

I appreciate the authors' efforts in the revision. My comments have been addressed.

Comments on the Quality of English Language

The text of the new-added Section 4.2.2.3 should be revised.

For example, the authors used imperative mood in the first sentence. The sentence "Change the regression method, and regress the simultaneous equation model on the above equation again" starts with an imperative form ("Change"), which is unusual for academic or technical writing. This could be rephrased for a more formal tone.

Also, the title "Table 8 Robustness test results 3" includes an extra "3" which seems out of place. This could be a typographical error.

Overall, the text has a few issues that impact its clarity and grammatical correctness. Rewriting the section will be suggested.

Author Response

We sincerely appreciate your recognition of our work and once again thank you for your professional and detailed review comments. We have carefully considered your feedback and made further revisions based on it. Specifically:

Feedback 1: The text of the newly added Section 4.2.2.3 should be revised. For example, the authors used the imperative mood in the first sentence. The sentence "Change the regression method, and regress the simultaneous equation model on the above equation again" starts with an imperative form ("Change"), which is unusual for academic or technical writing. This could be rephrased for a more formal tone.

Response: Thank you for your feedback. We agree that our previous wording was not precise enough, and we have rewritten the description of Section 4.2.2.3 accordingly.

Feedback 2: Also, the title "Table 8 Robustness test results 3" includes an extra "3" which seems out of place. This could be a typographical error. Overall, the text has a few issues that impact its clarity and grammatical correctness. Rewriting the section will be suggested.

Response: Thank you for your feedback. We had multiple "3s" in Table 8, which was mainly due to our incorporation of two additional robustness tests based on your suggestion. Currently, we have a total of three robustness tests in our article. To differentiate them, we added the numbers "1," "2," and "3" to the titles of each table. Upon your reminder, we realized that this could indeed be a confusing statement for readers. In the revised version, we have removed the extra numbers.

Once again, we appreciate the professional feedback you have provided for our article. Your input has greatly contributed to the improvement of our work. Wishing you a pleasant life.
